# Neutrino Spectrum and Energy Loss Rates Due to Weak Processes on Hot $^{56}$Fe in Pre-Supernova Environment

**A. A. Dzhioev** [1,*] ![ID], **A. V. Yudin** [2] ![ID], **N. V. Dunina-Barkovskaya** [2] and **A. I. Vdovin** [1]

[1]  Bogoliubov Laboratory of Theoretical Physics, Joint Institute for Nuclear Research, 141980 Dubna, Russia; vdovin@theor.jinr.ru

[2]  National Research Center "Kurchatov Institute", 123182 Moscow, Russia; yudin@itep.ru (A.V.Y.); dunina@itep.ru (N.V.D.-B.)

[*]  Correspondence: dzhioev@theor.jinr.ru; Tel.: +7-49621-63-385

**Abstract:** Applying TQRPA calculations of Gamow–Teller strength functions in hot nuclei, we compute the (anti)neutrino spectra and energy loss rates arising from weak processes on hot $^{56}$Fe under pre-supernova conditions. We use a realistic pre-supernova model calculated by the stellar evolution code MESA. Taking into account both charged and neutral current processes, we demonstrate that weak reactions with hot nuclei can produce high-energy (anti)neutrinos. We also show that, for hot nuclei, the energy loss via (anti)neutrino emission is significantly larger than that for nuclei in their ground state. It is found that the neutral current de-excitation via the $\nu\bar{\nu}$-pair emission is presumably a dominant source of antineutrinos. In accordance with other studies, we confirm that the so-called single-state approximation for neutrino spectra might fail under certain pre-supernova conditions.

**Keywords:** pre-supernova; hot nuclei; stellar evolution code MESA; (ant)neutrino spectra; (ant)neutrino energy loss rates

## 1. Introduction

It is well known that the production and propagation of (anti)neutrinos in the stellar matter are important ingredients of the computer modeling of stellar evolution. According to the theory, in stellar interiors with both high temperature and density, neutrino emission makes a major contribution to energy loss, removes entropy from the stellar core and accelerates the evolution of the star [1–3]. The observation of neutrinos from supernova SN1987A confirmed and advanced our understanding of core-collapse supernova explosion. Recent remarkable progress in neutrino detection techniques may enable the registration of neutrinos from new sources. Some of the candidates are pre-supernova (anti)neutrinos emitted from the core of a massive star just before the collapse [4]. Although pre-supernova (anti)neutrinos have not been detected to date, their observation would offer a possibility for studying the physical processes that lead to core collapse and would be a warning of an upcoming explosion.

In [5,6], the role of charged current nuclear weak processes (electron and positron capture, $\beta^{\mp}$-decay) in the neutrino emission from a pre-supernova star was studied. It was found that, under certain conditions, nuclear processes compete with thermal processes (plasmon decay, pair annihilation, etc.) in their contribution to the (anti)neutrino flux or even dominate in the energy window relevant for detection. However, it was pointed out that, while total emissivities are relatively robust, the highest-energy tails of the neutrino spectrum, in the detectable window, are very sensitive to the details of the calculations. Specifically, the source of the error lies in the single-strength approximation [7] that was adopted in [5,6] for nuclear processes. In [8], an exploratory study of this error was performed and it was shown that the specific neutrino spectrum obtained from the single-strength approximation could miss important features.

High-temperature stellar plasma allows nuclei to access excited states in accordance with the Boltzmann distribution. In [7], the single-strength approximation was derived assuming that (i) weak processes on a thermally excited state in the parent nucleus lead to the Gamow–Teller (GT) transition to a single state in the daughter nucleus and that (ii) the Brink hypothesis is valid, i.e., the GT strength function is the same for all excited states. The violation of the Brink hypothesis for thermally excited (hot) nuclei was demonstrated for both charge-exchange [9–11] and charge-neutral [12,13] Gamow–Teller strength functions, and it was shown that, under certain stellar conditions, thermal effects on the GT strength significantly affect the rates and cross-sections of the nuclear weak process (as can also be seen in recent reviews [14–16]).

In this paper, we apply the formalism of [9–16] to study electron (anti)neutrino spectra and energy loss rates arising from weak processes on hot $^{56}$Fe under conditions realized in the pre-supernova environment. Besides the charged current weak nuclear processes considered in [5,6], we also take into account the neutral current de-excitation of hot $^{56}$Fe via neutrino–antineutrino pair emission. The main goal of the present work is to study how thermal effects on the GT strength function and $\nu\bar{\nu}$-pair emission affects the (anti)neutrino spectra and energy loss rates.

## 2. Method

To compute (anti)neutrino spectra and energy loss rates due to weak processes on hot nuclei, we apply a method which is based on the statistical formulation of the nuclear many-body problem at finite temperature. In this method, rather than compute GT strength distributions for individual thermally excited states, we determine a thermal averaged strength function for the GT operator

$$S_{\mathrm{GT}_{\pm,0}}(E, T) = \sum_{i,f} p_i(T) B_{if}^{(\pm,0)} \delta(E - E_{if}),  \tag{1}$$

where $p_i(T) = e^{-E_i/kT}/Z(T)$ is the Boltzmann population factor for a parent state $i$ at a temperature $T$, $B_{if}^{\pm,0} = |\langle f \| \mathrm{GT}_{\pm,0} \| i \rangle|^2/(2J_i + 1)$ is the reduced transition probability (transition strength) from the state $i$ to the state $f$ in the daughter nucleus; $\mathrm{GT}_0 = \vec{\sigma} t_0$ for neutral current reactions and $\mathrm{GT}_{\mp} = \vec{\sigma} t_{\pm}$ for charged current reactions. The zero component of the isospin operator is denoted by $t_0$, while $t_-$ and $t_+$ are the isospin-lowering ($t_-|n\rangle = |p\rangle$) and isospin-rising ($t_+|p\rangle = |n\rangle$) operators. Thus, '0' refers to the $\nu\bar{\nu}$-pair emission, '−' to positron capture (PC) and $\beta^-$-decay, and '+' to electron capture (EC) and $\beta^+$-decay. The transition energy between initial and final states is given by $E_{if} = Q + E_f - E_i$, where $E_i$ and $E_f$ are the excitation energies of the parent and daughter nuclei, and $Q = M_f - M_i$ is the ground-state reaction threshold (for neutral current reactions $Q = 0$). The definition of $S_{\mathrm{GT}}(E, T)$ implies that at $T \neq 0$ the strength function is defined for both positive ($E > 0$) and negative ($E < 0$) energy domains. The latter corresponds to the de-excitation of thermally excited states to states at lower energies. In addition, low-energy transitions between excited states become possible at $T \neq 0$.

Obviously, the explicit state-by-state calculation of $S_{\mathrm{GT}_{\pm,0}}(E, T)$ is hardly possible due to the extremely large number of nuclear states thermally populated at stellar temperatures. To compute the temperature-dependent strength function (1), we apply the TQRPA framework which is a technique based on the quasiparticle random phase approximation (QRPA) extended to the finite temperature by the superoperator formalism in the Liouville space [14]. The central concept of the TQRPA framework is the thermal vacuum $|0(T)\rangle$, a pure state in the Liouville space, which corresponds to the grand canonical density matrix operator for the hot nucleus. The time-translation operator in the Liouville space is the so-called thermal Hamiltonian $\mathcal{H}$ constructed from the nuclear Hamiltonian after introducing particle creation and annihilation superoperators. Within the TQRPA, the strength func-

tion (1) is expressed in terms of the transition matrix elements from the thermal vacuum to eigenstates (thermal phonons) of the thermal Hamiltonian $\mathcal{H}|Q_i\rangle = \omega_i|Q_i\rangle$:

$$S_{\mathrm{GT}_{\pm,0}}(E,T) = \sum_i \mathcal{B}_i^{(\pm,0)} \delta(E - \omega_i \mp \Delta_{np}). \tag{2}$$

Here, $\mathcal{B}_i^{(\pm,0)} = |\langle Q_i\|\mathrm{GT}_{\pm,0}\|0(T)\rangle|^2$ is the transition strength to the $i$th state of a hot nucleus and $E_i^{(\pm,0)} = \omega_i \pm \Delta_{np}$ is the transition energy; $\Delta_{np} = 0$ for charge-neutral transitions, while for charge-exchange transitions $\Delta_{np} = \delta\lambda_{np} + \delta M_{np}$, where $\delta\lambda_{np} = \lambda_n - \lambda_p$ is the difference between neutron and proton chemical potentials in the nucleus, and $\delta M_{np} = 1.293\,\mathrm{MeV}$ is the neutron–proton mass splitting. Note that the eigenvalues of the thermal Hamiltonian, $\omega_i$, take both positive and negative values. The latter contribute to the strength function only at $T \neq 0$. We also stress that the strength function (2) obeys the detailed balance principle:

$$S_{\mathrm{GT}_0}(-E,T) = \mathrm{e}^{-E/kT} S_{\mathrm{GT}_0}(E,T) \tag{3}$$

for charge-neutral GT transitions, and

$$S_{\mathrm{GT}_\mp}(-E,T) = \mathrm{e}^{-(E\mp\Delta_{np})/kT} S_{\mathrm{GT}_\pm}(E,T) \tag{4}$$

for charge-exchange GT transitions. This property makes the approach thermodynamically consistent.

In what follows, we assume that emitted (anti)neutrinos freely leave the star. Then, we can write the following expressions for electron (anti)neutrino spectra resulting from the GT transition from the thermal vacuum to the $i$th state of a hot nucleus:

- Electron or positron capture

$$\lambda_i^{\mathrm{EC,\,PC}}(E_\nu) = \frac{G_\mathrm{F}^2 V_{\mathrm{ud}}^2 (g_A^*)^2}{2\pi^3\hbar^7 c^6} \mathcal{B}_i^{(\pm)} (E_\nu + E_i^{(\pm)})[(E_\nu + E_i^{(\pm)})^2 - m_e^2 c^4]^{1/2} E_\nu^2$$
$$\times f_{e^\mp}(E_\nu + E_i^{(\pm)}) F(\pm Z, E_\nu + E_i^{(\pm)}) \Theta(E_\nu + E_i^{(\pm)} - m_e c^2), \quad (5)$$

  where upper (lower) sign corresponds to EC (PC);
- $\beta^\mp$-decay

$$\lambda_i^{\beta^\mp}(E_\nu) = \frac{G_\mathrm{F}^2 V_{\mathrm{ud}}^2 (g_A^*)^2}{2\pi^3\hbar^7 c^6} \mathcal{B}_i^{(\mp)} (-E_\nu - E_i^{(\mp)})[(-E_\nu - E_i^{(\mp)})^2 - m_e^2 c^4]^{1/2} E_\nu^2$$
$$\times [1 - f_{e^\mp}(-E_\nu - E_i^{(\mp)})] F(\pm Z + 1, -E_\nu - E_i^{(\mp)}) \Theta(-E_\nu - E_i^{(\mp)} - m_e c^2), \quad (6)$$

  where upper (lower) sign corresponds to $\beta^-$- ($\beta^+$-)decay;
- $\nu\bar\nu$-pair emission produces the same spectra for $\nu_e$ and $\bar\nu_e$ (The spectrum of other (anti)neutrino flavors is also given by (7).)

$$\lambda_i^{\nu\bar\nu}(E_\nu) = \frac{G_\mathrm{F}^2 g_A^2}{2\pi^3\hbar^7 c^6} \mathcal{B}_i^{(0)} (-E_\nu - E_i^{(0)})^2 E_\nu^2 \Theta(-E_\nu - E_i^{(0)}). \tag{7}$$

In the above expressions, $G_\mathrm{F}$ denotes the Fermi coupling constant, $V_{\mathrm{ud}}$ is the up–down element of the Cabibbo–Kobayashi–Maskava quark-mixing matrix and $g_A = -1.27$ is the weak axial coupling constant. Note that, for charged current reactions, we use the effective coupling constant $g_A^* = 0.74 g_A$ that takes into account the observed quenching of the $\mathrm{GT}_\pm$ strength. The function $f_{e^-(e^+)}(E)$ is the Fermi–Dirac distribution for electrons (positrons), and the Fermi function $F(Z, E)$ takes the distortion of the charged lepton wave function by the Coulomb field of the nucleus into account. It follows from the energy conservation that, for capture reactions, the electron (positron) energy is given by $E_{e^\mp} = E_\nu + E_i^{(\pm)}$, while for the $\beta^\pm$-decay, we have $E_{e^\mp} + E_\nu = -E_i^{(\mp)}$, and $E_{\bar\nu} + E_\nu = -E_i^{(0)}$ for $\nu\bar\nu$-pair

emission. Obviously, only negative-energy transitions ($E_i^{(\pm,0)} < 0$) contribute to $\beta^{\mp}$-decay and $\nu\bar{\nu}$-pair emission.

Summation over different contributions x = EC, $\beta^+$, $\nu\bar{\nu}$ (PC, $\beta^-$, $\nu\bar{\nu}$) and final states $i$ of a hot nucleus gives us the total (anti)neutrino spectrum

$$\lambda(E_\nu) = \sum_x \sum_i \lambda_i^x(E_\nu). \tag{8}$$

Then, the integration over $E_\nu$ yields the neutrino emission ($\Lambda$) and energy-loss ($P$) rates

$$\Lambda = \int \lambda(E_\nu)\, dE_\nu, \quad P = \int \lambda(E_\nu) E_\nu dE_\nu. \tag{9}$$

## 3. Results

### 3.1. Pre-Supernova Model

To study (anti)neutrino production and energy loss rates due to hot $^{56}$Fe in the pre-supernova environment, we use the model `25_79_0p005_ml` from Farmer et al. [17]. It is a typical pre-supernova model with a good mass resolution and a core temperature that is high enough for our estimates. Its name means that the initial mass of the model was $25 M_\odot$, the nuclear reaction network was `mesa_79.net`, the maximum mass of a computational cell was $0.005 M_\odot$, and the mass loss during the stellar evolution was taken into account (see details in [17]).

The authors of [17] employed the stellar evolution code MESA [18], version 7624. In output, MESA gives the time-evolving profiles of density $\rho$ (in g/ccm), temperature $T_9 \equiv T(K)/10^9$, electron fraction $Y_e$ and mass fraction $X_i$ of various isotopes. The profile that we use corresponds to the onset of core collapse, which is defined as the time when the infall velocity exceeds 1000 km/s anywhere in the star. The respective density, temperature and electron fraction profiles along the mass coordinate are demonstrated in the top panels of Figure 1. In the bottom panel of Figure 1, we show the mass fraction profiles of the most dominant isotopes. We see that the $^{56}$Fe isotope is dominant up to $m < 1.3 M_\odot$. It is in this hot and dense central part of the star that the main neutrino flux is born.

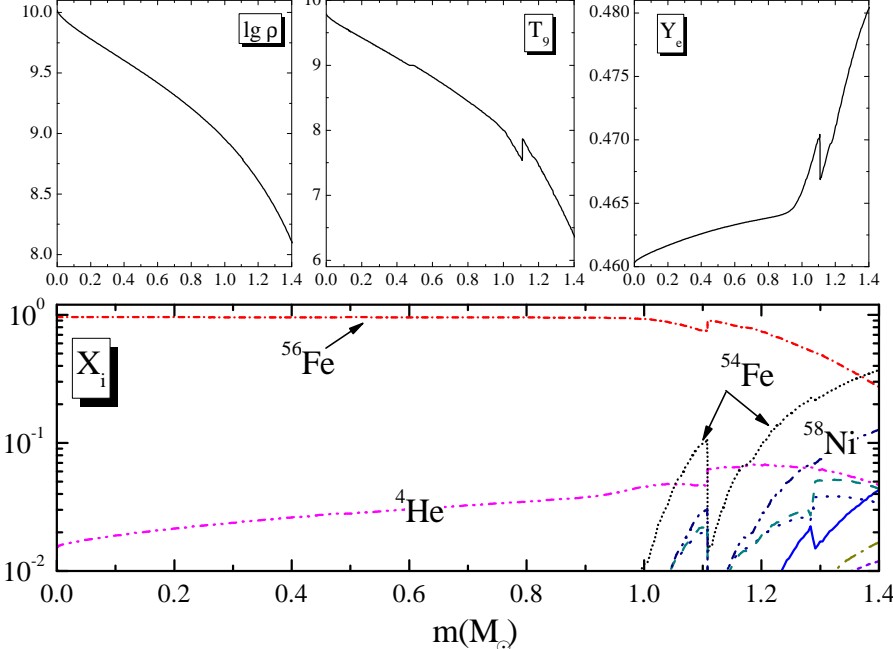

**Figure 1.** **Top** panels: density, temperature and electron fraction profiles along the mass coordinate for the `25_79_0p005_ml` pre-supernova model at the onset of the core collapse. **Bottom** panel: the respective mass fraction distribution of the most dominant isotopes.

We calculate (anti)neutrino spectra and energy loss rates due to hot $^{56}$Fe at six specific points on the mass coordinate for which $X_{^{56}\text{Fe}} > 0.5$. To select these points, in the MESA output file, we first identify the mass coordinate $m_{(6)}$ where $X_{^{56}\text{Fe}}$ takes the value closest to 0.5. Then, the remaining five points are taken from the MESA output and are uniformly distributed over the interval $[0, m_{(6)}]$. The values of $m_{(n)}$ with the respective values of the radial coordinate, $^{56}$Fe mass fraction, density, temperature, electron fraction and electron chemical potential are given in Table 1. It is clearly seen from Table 1 that the range of temperature and density varies widely, while the electron fraction remains almost unchanged. The resulting chemical potential reduces four times when we move along the mass coordinate from point $m_{(1)}$ to $m_{(6)}$. Thus, the selected points enable us to consider weak nuclear processes under rather different representative pre-supernova conditions.

**Table 1.** Six specific points on the mass coordinate where the (anti)neutrino spectra and energy loss rates due to hot $^{56}$Fe are computed.

| (n) | $m$ ($M_\odot$) | $R$ ($R_\odot$) | $X_{^{56}\text{Fe}}$ | $T_9$ | $\log(\rho)$ | $Y_e$ | $\mu_e$ (MeV) [1] |
|---|---|---|---|---|---|---|---|
| (1) | $1.953 \times 10^{-6}$ | $6.41 \times 10^{-6}$ | 0.95822 | 9.79138 | 10.01954 | 0.46029 | 8.451 |
| (2) | 0.26005 | $3.75 \times 10^{-4}$ | 0.95791 | 9.33784 | 9.72765 | 0.46197 | 6.679 |
| (3) | 0.51745 | $5.22 \times 10^{-4}$ | 0.95872 | 8.95570 | 9.49678 | 0.46304 | 5.527 |
| (4) | 0.77778 | $6.71 \times 10^{-4}$ | 0.95356 | 8.49247 | 9.23424 | 0.46379 | 4.435 |
| (5) | 1.03597 | $8.51 \times 10^{-4}$ | 0.88566 | 7.86298 | 8.90200 | 0.46725 | 3.340 |
| (6) | 1.29568 | $1.13 \times 10^{-3}$ | 0.497 | 6.97018 | 8.39942 | 0.47616 | 2.126 |

[1] The chemical potential $\mu_e$ is defined to include the rest mass so that $\mu_{e^-} = -\mu_{e^+}$. The value of $\mu_e$ is determined from the density $\rho Y_e$ by inverting the relation $\rho Y_e = (\pi^2 \hbar^3 N_A)^{-1} \int_0^\infty (f_{e^-} - f_{e^+}) p^2 dp$.

### 3.2. Thermal Effects on Gamow–Teller Strength Functions in $^{56}$Fe

Before discussing (anti)neutrino spectra and energy loss rates, we consider the thermal evolution of the GT strength functions in $^{56}$Fe. In Figure 2, the $\text{GT}_{0,\mp}$ strength functions are displayed at three temperatures relevant in the pre-supernova context. To emphasize thermal effects, the ground-state strength functions are also shown in each panel. The choice of the nuclear model and its parameters for TQRPA calculations is discussed in [15]. Here, we just mention that the strength functions in Figure 2 are obtained by applying self-consistent calculations based on the SkM* parametrization of the Skyrme effective force. As shown in [15], zero-temperature QRPA calculations with the SkM* force fairly accurately reproduce both experimental data and shell-model results on the $\text{GT}_{0,\mp}$ resonance in the ground state of $^{56}$Fe (TQRPA calculations performed with Skyrme forces SLy4, SkO' and SGII [9–16] clearly demonstrate that thermal effects on the GT strength functions do not depend on a particular choice of the parametrization—for this reason, all the results presented below concerning the temperature dependence of (anti)neutrino spectra and energy loss rates remain valid for other Skyrme parametrizations.). According to the present calculations, the main contribution to the $\text{GT}_0$ resonance ($E \approx 15\,\text{MeV}$) in $^{56}$Fe comes from proton and neutron charge-neutral single-particle transitions $1f_{7/2} \to 1f_{5/2}$, while the $\text{GT}_-$ and $\text{GT}_+$ resonances at energies of $E \approx 15\,\text{MeV}$ and $E \approx 6\,\text{MeV}$, respectively, are mainly formed by the $1f_{7/2} \to 1f_{5/2}$ charge-exchange transitions.

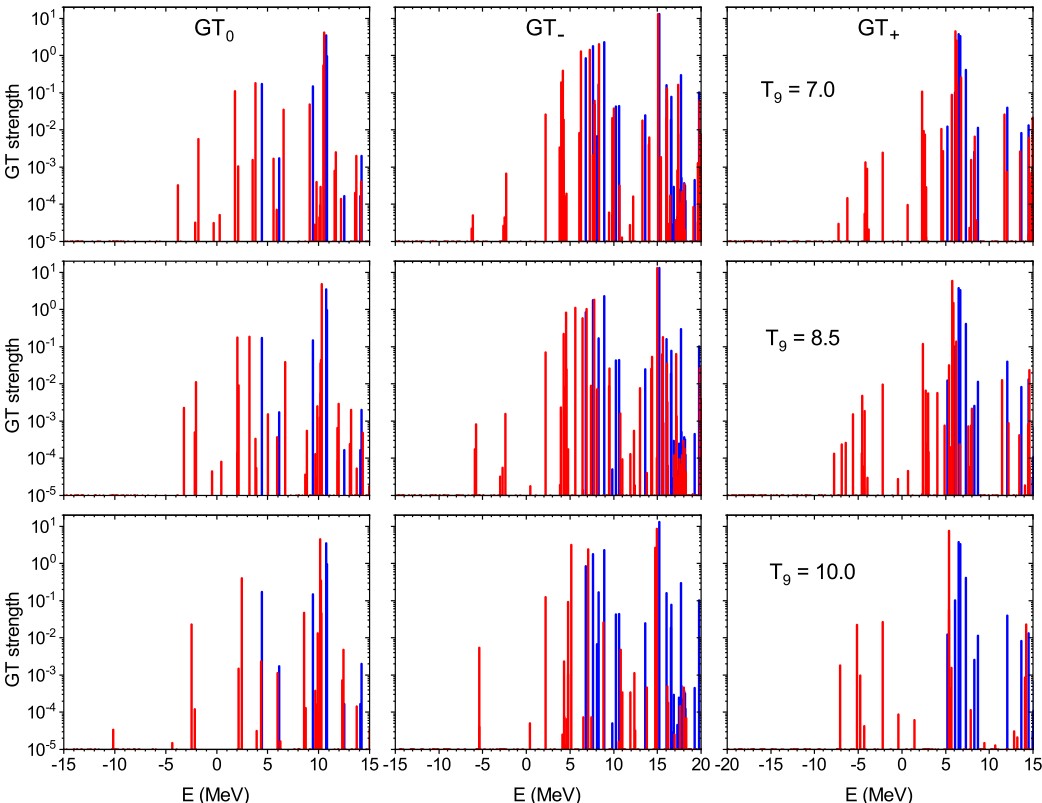

**Figure 2.** $GT_0$ (**left** column), $GT_-$ (**middle** column) and $GT_+$ (**right** column) strength functions $S_{GT}$ for $^{56}$Fe calculated at $T_9 = 7.0$ (**upper** row), $T_9 = 8.5$ (**middle** row) and $T_9 = 10.0$ (**lower** row). The blue bars represent the ground-state strength functions.

As seen from the plots, the thermal effects can noticeably change the strength functions. First, since our TQRPA calculations do not support the Brink hypothesis, the GT strength for upward ($E > 0$) transitions exhibits a temperature dependence. Namely, due to the vanishing of pairing correlations and thermal weakening of the residual particle–hole interaction, the $GT_{0,\mp}$ resonance moves to lower energies. Moreover, the thermal smearing of the nuclear Fermi surfaces unblocks the low-energy GT transitions. In charge-exchange strength functions $S_{GT_-}$ and $S_{GT_+}$, these transitions lead to the appearance of the GT strength below the ground-state reaction threshold $Q$ (for $^{56}$Fe $\rightarrow$ $^{56}$Mn reactions $Q = 4.207$ MeV, and for $^{56}$Fe $\rightarrow$ $^{56}$Co reactions $Q = 4.055$ MeV) , while in the $GT_0$ distribution, finite temperature unblocks a low-energy strength below the experimental energy of the first $1^+$ state in $^{56}$Fe ($E_{1_1^+} \approx 3.12$ MeV). Second, a temperature rise increases the population of nuclear excited states and enables downward ($E < 0$) transitions in accordance with the detailed balance relations (3) and (4). Comparing the $GT_-$ and $GT_+$ distributions at $T \neq 0$, we see that the main contribution to the negative-energy $GT_-$ strength comes from the transition, which is inverse to the $GT_+$ resonance. At the same time, the main contribution to the $GT_+$ strength at $E < 0$ comes from transitions inverse to low-energy $GT_-$ transitions, while the contribution of the transition inverse to the $GT_-$ resonance is small. The reason for this is that the $GT_-$ resonance is much higher in energy than the $GT_+$ resonance, and therefore its inverse transition is strongly suppressed by the Boltzmann exponential factor in the detailed balance relation (4). In the $GT_0$ distribution, negative-energy transitions inverse to low-energy ones and to the excitation of the $GT_0$ resonance contribute to the downward strength.

It is important to emphasize that, since upward and downward strengths are connected by the detailed balance relation, thermal effects on the upward GT strength also influence the downward strength. In [12], this influence was studied by comparing the running (cumulative) sums for the $GT_0$ downward strength calculated using and without using the Brink hypothesis. In particular, it was shown that both the thermal unblocking of

low-energy strength and lowering the GT resonance significantly enhance the strength of negative-energy transitions. Eventually, this enhancement should have important consequences for (anti)neutrinos emitted due de-excitation and decay processes.

### 3.3. (Anti)neutrino Spectra and Energy Loss Rates

We now demonstrate the $\nu_e$ and $\bar{\nu}_e$ energy spectra for six selected points inside the star. Figure 3 shows the contribution of different nuclear weak processes to neutrino spectra for each point from Table 1. Although the shape and intensity of the spectrum depend on the temperature, density and electron fraction, there are features common for all points. Namely, for all points, the spectra are dominated by the EC contribution that exhibits a low-energy peak and a high-energy tail. The latter gradually transforms into the second peak when we move from the center of the star. Our analysis shows that low-energy neutrinos are emitted after electron capture excites the $GT_+$ resonance state, while high-energy neutrinos are caused by thermally unblocked low- and negative-energy $GT_+$ transitions.

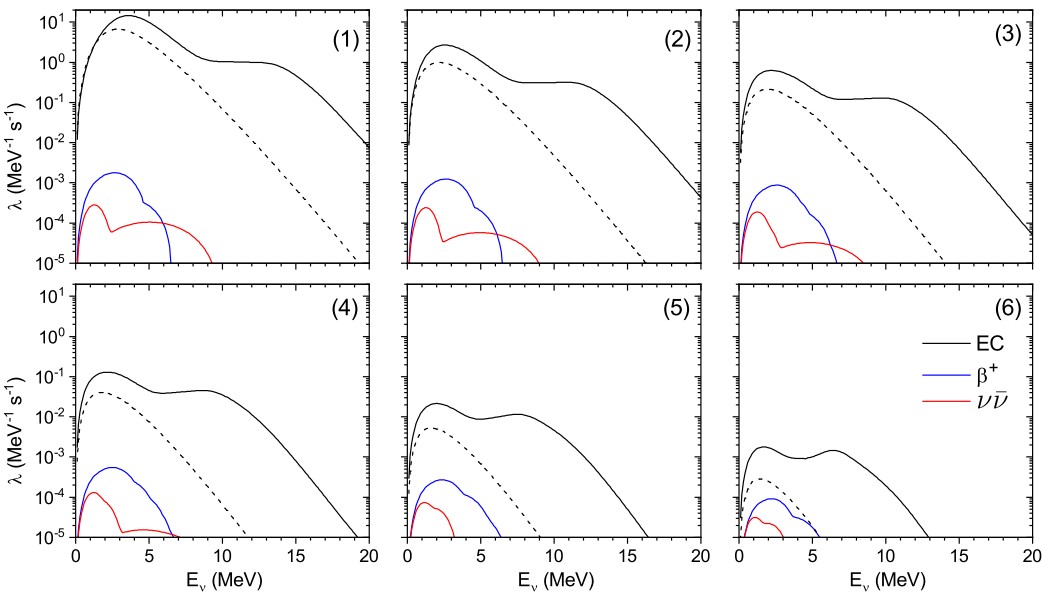

**Figure 3.** Neutrino spectra produced by $^{56}$Fe due to electron capture (EC), $\beta^+$-decay and $\nu\bar{\nu}$-pair emission. Each set of curves corresponds to a specific point (n) (n = 1, 2, 3, 4, 5, 6) on the mass coordinate listed in Table 1. The dashed curves represent neutrino spectra arising from EC on the ground state of $^{56}$Fe.

In Figure 3, the importance of thermal effects is illustrated by comparing neutrino spectra produced by hot $^{56}$Fe with that produced by a cold nucleus, when only EC is possible. As clearly seen from Figure 3, the thermal unblocking of the $GT_+$ strength at $E < 0$ (see the right panels in Figure 2) leads to the appearance of high-energy neutrinos in the spectra, whose fraction increases when we move from point (1) to (6). Moreover, as shown in the figure, the temperature-induced lowering of the $GT_+$ resonance amplifies the low-energy ($E_{\nu_e} < 5$ MeV) part of the spectra and shifts its maximum to higher energies.

The contribution of different nuclear weak processes to the antineutrino spectra produced by hot $^{56}$Fe is shown in Figure 4. Our calculations clearly demonstrate the dominance of $\nu_e\bar{\nu}_e$-pair emission in the antineutrino spectra under pre-supernova conditions when the $\beta^-$-decay is strongly blocked by the electron chemical potential. The obtained $\nu_e\bar{\nu}_e$-spectra have a narrow low-energy peak at $E_\nu \approx 1$–2 MeV and a broad high-energy one peaking around $E_\nu \approx 5$ MeV. By matching with the $GT_0$ strength function in Figure 2, we conclude that the former arises due to thermally unblocked low-energy downward $GT_0$ transitions, while high-energy antineutrinos are emitted from the $\nu_e\bar{\nu}_e$-decay of the thermally populated $GT_0$ resonance. Since the $GT_0$ resonance in $^{56}$Fe is located at relatively high energy, its

thermal population rapidly decreases at low temperatures, leading to a decrease in the fraction of high-energy antineutrinos. Nevertheless, amongst weak nuclear processes, it is the $\nu_e\bar{\nu}_e$-decay of the $GT_0$ resonance that produces the high-energy antineutrinos of all flavors under pre-supernova conditions listed in Table 1. It is also seen from Figure 4 that temperature reduction has a modest impact on the intensity of low-energy antineutrinos emitted due to the $\nu_e\bar{\nu}_e$-decay. At the same time, the reduction in the chemical potential $\mu_e$ unblocks $\beta^-$-decay, which also emits low-energy antineutrinos. As a result, when we move from the center of the star, the contributions of the $\nu_e\bar{\nu}_e$-pair emission and $\beta^-$-decay to the low-energy antineutrino spectrum become comparable.

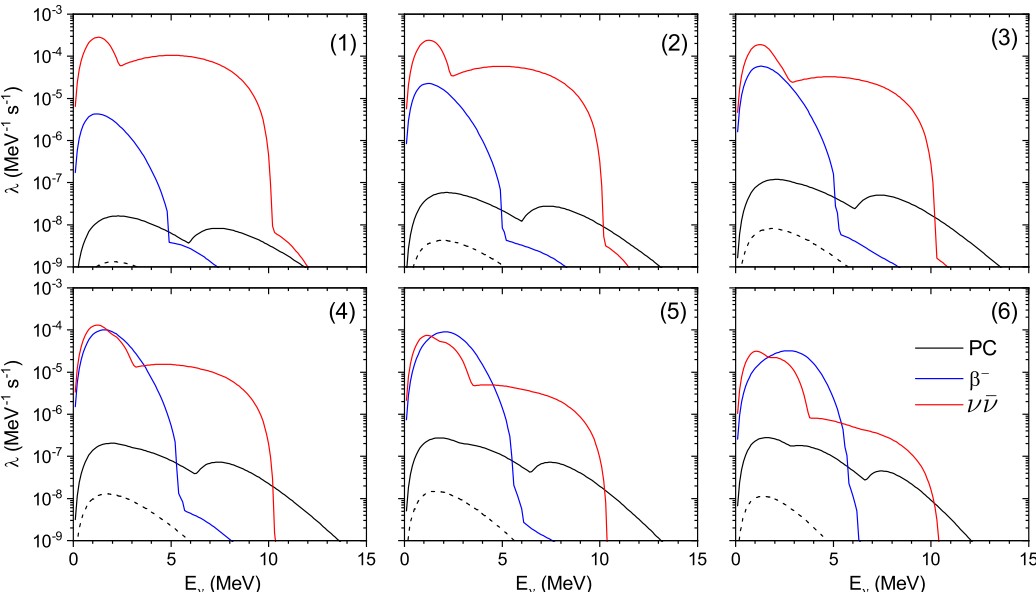

**Figure 4.** Antineutrino spectra produced by $^{56}$Fe due to positron capture (PC), $\beta^-$-decay and $\nu\bar{\nu}$-pair emission. Each set of curves corresponds to a specific point (n) (n = 1, 2, 3, 4, 5, 6) on the mass coordinate listed in Table 1. The dashed curves represent antineutrino spectra arising from PC on the ground state of $^{56}$Fe.

Figure 5 shows the evolution of the total (anti)neutrino spectrum $\lambda$ (8) as we move from the center of the star. Since electron capture is a dominant source of neutrinos, the reduction in the chemical potential $\mu_e$ below the $GT_+$ resonance energy decreases the low-energy peaks in $\lambda$ by about four orders of magnitude, while the high-energy tail is reduced by approximately three orders of magnitude. For this reason, a relative fraction of high-energy neutrinos in the spectrum increases. As discussed above, contributions from the $\nu\bar{\nu}$-pair emission and $\beta^-$-decay to emission of low-energy antineutrinos demonstrate opposite trends when we move from points (1) to (6). Therefore, the intensity of the low-energy antineutrino emission is rather unsensitive to the change in pre-supernova conditions. At the same time, the intensity of high-energy antineutrino emission is reduced by more than two orders of magnitude as the temperature decreases from $T_9 \approx 9.8$ to $T_9 \approx 7.0$.

In Figure 6, the emission rates $\Lambda$, energy-loss rates $P$, and the average energy $\langle E_\nu \rangle = P/\Lambda$ for the electron (anti)neutrinos emitted due to weak processes with hot $^{56}$Fe are shown. Referring to the figure, neutrino rates demonstrate a strong dependence under pre-supernova conditions. Compared with the ground-state rates, we conclude that temperature lowering gives a minor contribution to a severe reduction in the neutrino rates and the latter is mainly caused by the chemical potential decrease. In contrast, as pair emission only depends on temperature and $\beta^-$-decay rate increases when $\mu_e$ decreases, the computed antineutrino rates demonstrate a more modest dependence under pre-supernova conditions. We also see that the finite temperature of the nucleus plays a more important role for antineutrino rates than for neutrino ones.

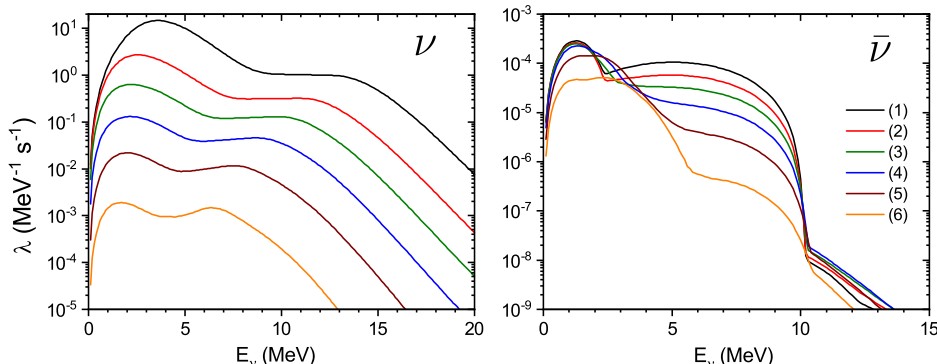

**Figure 5.** Neutrino and antineutrino spectra $\lambda$ due to weak processes with hot $^{56}$Fe for specific points (n) (n = 1, 2, 3, 4, 5, 6) on the mass coordinate listed in Table 1.

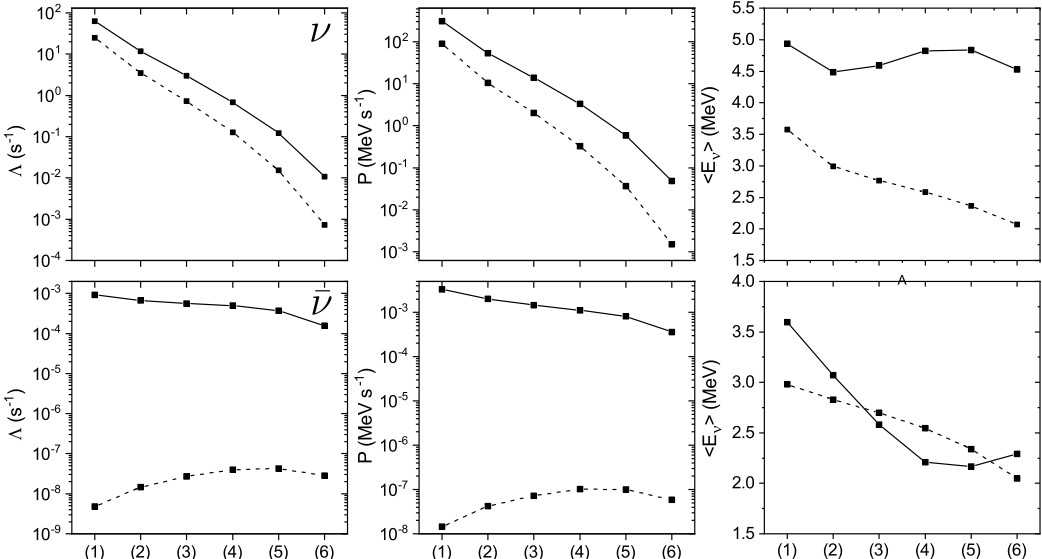

**Figure 6.** Neutrino (**top**) and antineutrino (**bottom**) emission rate $\Lambda$, energy-loss rate $P$ and average energy $\langle E_\nu \rangle$ due to hot $^{56}$Fe for specific points (n) (n = 1, 2, 3, 4, 5, 6) on the mass coordinate listed in Table 1. The dashed curves show $\Lambda$, $P$, and $\langle E_\nu \rangle$ calculated for the ground state of $^{56}$Fe.

As for the average energy, for emitted neutrinos, it varies rather weakly around $\langle E_\nu \rangle \approx 4.7$ MeV. This stability is a result of the increasing fraction of high-energy neutrinos emitted by de-excitation processes, which compensates the decrease in available electron energy when we move from the center of the star. This is clearly seen if we compute $\langle E_\nu \rangle$ for the cold $^{56}$Fe. In that case, $\langle E_\nu \rangle$ is essentially lower and shows a decreasing trend. At the same time, the average energy of antineutrinos demonstrates non-monotonic behavior due to the competition between $\nu\bar{\nu}$-decay and $\beta^-$-decay. Moreover, since, in decay processes, the released energy is shared among two emitted particles, the antineutrino average energy is smaller than that for neutrinos.

## 4. Discussion and Perspectives

Neutrino spectra shown in Figure 3 confirm the conclusion of Ref. [7] that the single-strength approximation can be applied under stellar conditions with the electron chemical potential high enough to allow the excitation of the $GT_+$ resonance by electron capture. Such conditions occur during the collapse phase. However, our calculations clearly demonstrate that this approximation can fail in the pre-supernova phase when negative-energy $GT_+$ transitions from thermally excited states noticeably contribute to electron capture and the resulting neutrino energy spectrum is double-peaked. On the whole, the present thermodynamically consistent calculations of electron neutrino spectra performed without

assuming the Brink hypothesis indicate that the thermal effects on the $GT_+$ strength function shift the spectrum to higher energies, and thus make the neutrino detection more likely.

The inclusion of $\nu\bar{\nu}$-pair emission into consideration shows that this neutral current process might be a dominant source of high-energy antineutrinos emitted via the de-excitation of the $GT_0$ resonance. Considering that the energy of the $GT_0$ resonance is related to the spin-orbit splitting, the high-energy peak in antineutrino spectra can be easily parameterized. Moreover, since the $\nu\bar{\nu}$-pair emission only depends on temperature, the detection of high-energy pre-supernova antineutrinos might be a test for thermodynamic conditions in the stellar interior.

The next evident step in our study of the role of nuclear weak processes in pre-supernova (anti)neutrino production is to compute overall (anti)neutrino spectra and energy loss rates as well as their time evolution for different stellar progenitors. To this end, calculations such as those performed for $^{56}$Fe are needed for isotopes abundant in the stellar core and then, in the integration over the whole core, these should be performed for several time steps. Concerning the possibility of (anti)neutrino detection, we should take into account (ant)neutrino flavor oscillation, which changes the initial flavor composition of the pre-supernova (anti)neutrino flux.

**Author Contributions:** Conceptualization: A.A.D. and A.V.Y.; formal analysis: A.A.D., A.V.Y., N.V.D.-B. and A.I.V.; software: A.A.D., A.V.Y. and N.V.D.-B.; writing—original draft preparation: A.A.D.; writing—review and editing: A.A.D., A.I.V., A.V.Y. and N.V.D.-B. All authors have read and agreed to the published version of the manuscript.

**Funding:** A.V.Y. thanks RSF 21-12-00061 grant for support.

**Data Availability Statement:** Not applicable.

**Conflicts of Interest:** The authors declare no conflict of interest.

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
