# Peer review of "Neutrino Spectrum and Energy Loss Rates Due to Weak Processes on Hot 56Fe in Pre-Supernova Environment"

_2571-712X, doi:10.3390/particles6030041_

Round 1

Reviewer 1 Report

This manuscript presents the results of a study of the neutrino spectrum and energy loss rates on "hot" 56Fe in the pre-supernova environment. It is of interest to the community engaged in similar studies.  Some agreements with past work are detailed, as well as some important differences. The manuscript is very well written and, in my opinion, certainly deserves to be published.

Author Response

Reviewer does not require any revision. 

We are grateful the reviewer for a possitive responce.

Reviewer 2 Report

In this paper the authors applied so-called thermal quasiparticle random-phase approximation (TQRPA) for calculations of Gamow-Teller strength functions in hot nuclei, and to compute (anti)neutrino spectra and energy loss rates arising from weak processes on hot 56Fe at pre-supernova conditions. A realistic pre-supernova model calculated by the stellar evolution code MESA is used. In the calculations are taken into account both charged and neutral current processes. The authors showed that weak reactions with hot nuclei can produce high-energy (anti)neutrinos and that for hot nuclei the energy loss via (anti)neutrino emission is significantly larger than that for nuclei in their ground state. It is found that neutral current de-excitation via $\nu\overline{\nu}$-pair emission is presumably a dominant source of antineutrinos. The authors indicated opportunities for future research at the end of the paper.

As it is written in lines 149-150: "Here we just mention that the strength functions in Figure 2 are obtained applying self-consistent calculations based on the SkM* parametrization of the Skyrme effective force." It would be good if the authors discuss how different parameterizations of the Skyrme effective force will be affected on the results given in the paper. 

Summarizing. I think this paper is interesting and valuable contribution to the field. The figures and tables are very informative. The paper is well written and articulated, and once the point raised above are addressed the manuscript can be accepted for publication in Particles. 

Author Response

We are grateful to the reviewer for a positive response and a valuable comment.

On the reviewer comment

“It would be good if the authors discuss how different parameterizations of the Skyrme effective force will be affected on the results given in the paper.”

we add the following reply in line 153 as a footnote

“TQRPA calculations performed with Skyrme forces SLy4, SkO', SGII [9-16] clearly demonstrate that thermal effects on the GT strength functions do not depend on a particular choice of the parametrization. For this reason all the results presented below concerning temperature dependence of (anti)neutrino spectra and energy loss rates  remain valid for other Skyrme parametrizations.”